# Carbonaceous Nanocomposites for Biomedical Applications as High-Drug Loading Nanocarriers for Sustained Delivery: A Review

**Bo Sun [1,2,3], Weijun Wang [2] and Mohini Sain [1,*]**

[1] Center for Biocomposites and Biomaterials Processing, Department of Mechanical and Industrial Engineering, University of Toronto, 33 Willcocks St., Toronto, ON M5S 3B3, Canada

[2] Key Laboratory of Food Nutrition and Safety, Tianjin University of Science and Technology, Ministry of Education, Tianjin 300457, China

[3] Department of Chemical Engineering, University of New Brunswick, Fredericton, NB E3B 5A3, Canada

[*] Correspondence: m.sain@utoronto.ca; Tel.: +1-416-946-3191; Fax: +1-416-978-3834

**Abstract:** Low drug loading and high initial burst release are common drawbacks for most polymeric nanocarriers in their biomedical applications. This review emphasizes the use of unconventional carbonaceous nanocomposites as functional carriers to improve the drug loading capacity and their capability of protecting drugs from the surrounding environment. The unique properties of typical carbonaceous nanocarriers, including nanotube, graphene/graphite, fullerene, and nanodiamonds/diamond-like carbon, are presented. Advanced methods for the surface functionalization of carbonaceous nanocarriers are described, followed by a summary of the most appealing demonstrations for their efficient drug loading and sustained release in vitro or in vivo. The fundamental drug delivery concepts based on controlling mechanisms, such as targeting and stimulation with pH, chemical interactions, and photothermal induction, are discussed. Additionally, the challenges involved in the full utilization of carbonaceous nanocomposites are described, along with the future perspectives of their use for enhanced drug delivery. Finally, despite its recent emergence as a drug carrier, carbon-based nanocellulose has been viewed as another promising candidate. Its structural geometry and unique application in the biomedical field are particularly discussed. This paper, for the first time, taxonomizes nanocellulose as a carbon-based carrier and compares its drug delivery capacities with other nanocarbons. The outcome of this review is expected to open up new horizons of carbonaceous nanocomposites to inspire broader interests across multiple disciplines.

**Keywords:** carbonaceous nanocomposites; biomedical application; high drug loading; sustained release; surface functionalization; carbon-based nanocellulose

## 1. Introduction

Most of the existing polymeric nanocarriers possess low drug loading capacity (generally less than 10%) towards biomedical applications [1]. This will lead to repeated drug administration and high treatment costs. Correspondingly, the side effects of the drug would be multiplied and may cause permanent health damage, or even life-threatening syndromes. Therefore, drug loading efficiency remains an essential component in the design of drug carriers. The short circulation half-life of these polymeric nanocarriers causes their faster elimination via opsonization by phagocytes inside the human body, which substantially constrains their application for sustained drug delivery [2]. As a solution to these problems, carbonaceous nanocomposites have emerged as a booming technique in the biomedical field to enhance the drug loading/release capacity.

Carbonaceous nanocomposites refer to a family of functional composites made up of carbon nanomaterials, which can be used in a broad range of applications due to their unique physical–chemical properties. As for the purpose of drug delivery, the internal

structure, morphology, and surface chemistry of carbon nanoparticles are the key factors that greatly affect their reaction with drugs. Typical carbonaceous nanocomposites possess a network structure of ordered nanochannels and high surface area, which allow more drugs to be loaded with a faster adsorption rate. Numerous works have been documented in the literature which demonstrate the function of these mesoporous carbonaceous materials in serving as carriers for sustained and high-loading drug delivery [3–6].

Among all the active carbonaceous carriers, nanotube, graphene/graphite, fullerene, and nanodiamonds/diamond-like carbon are the most extensively studied for drug delivery, largely due to their tunable physicochemical properties and facile surface functionalization [7]. Shown in Figure 1 are the schematic structures of various carbonaceous derivatives, which are formulated in different dimensions. More recently, the biomass-based renewable carbon nanocellulose has also been proposed as an advanced carrier for drug loading due to its extraordinary geometry and superb biocompatibility. In this review, we present an overview of recent advances in these carbonaceous nanocomposites for their drug delivery use. The critical issues that need to be addressed for the full utilization of carbonaceous nanocomposites in biomedical fields are discussed as well.

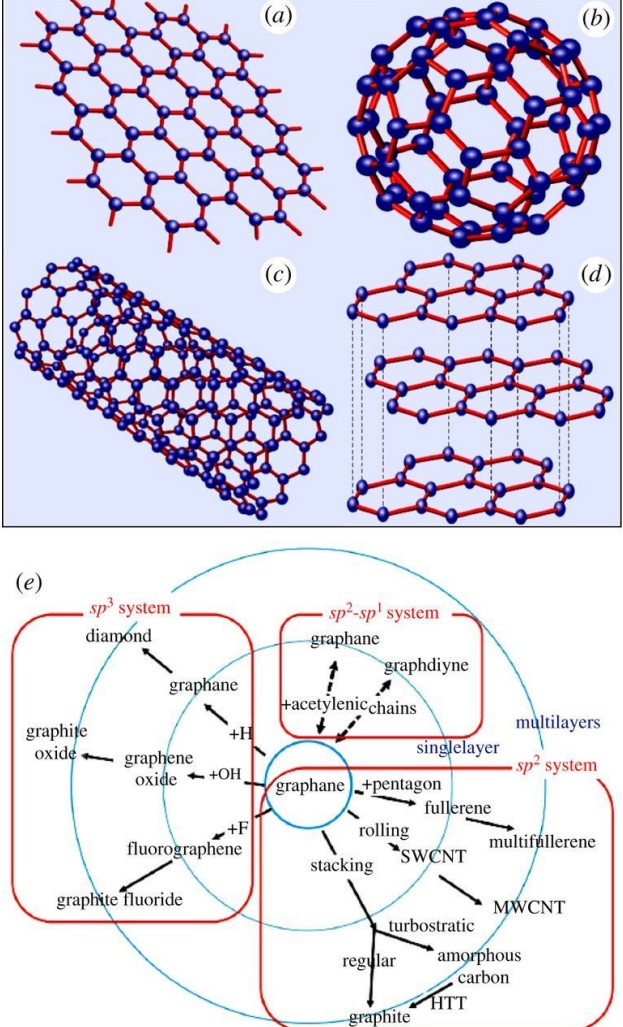

**Figure 1.** Schematic structures of (**a**) graphene (2 dimensions), (**b**) fullerene molecule (0 dimensions), (**c**) carbon nanotubes (1 dimension), (**d**) graphite (3 dimensions); (**e**) schematic of different carbonaceous derivatives including diamond [8].

## 2. Carbon Nanotubes (CNTs)

Carbon nanotubes (CNTs) have been widely used as functional carrier vectors with high loading capacity for sustained drug delivery. CNTs are cylinder-shaped allotropes of carbon, which can be categorized into two major genres, single-walled carbon nanotubes (SWCNTs) and multi-walled carbon nanotubes (MWCNTs), based on the number of concentric layers and wrapping mode. SWCNTs are present in stiff, rod-like tubes due to the increased van der Waals force, resulting from their nano-scaled diameter (internal diameter of approximately 1 nm) and high surface area (theoretically estimated between 50 and 1315 $m^2/g$), which have enhanced loading capability. A previous study revealed the poly(ethylene glycol) (PEG) functionalized SWNTs could load as high as 400% of doxorubicin (DOX) by weight [9]. In contrast to SWCNTs, MWCNTs have a larger internal diameter of 5–20 nm with a curly and agglomerated structure. They are partially dispersible in water, and can form translucent dispersions, whereas SWCNTs are mostly insoluble and prone to aggregate in an aqueous environment. Both SWCNTs and MWCNTs have high propensity towards functional modification, which can further strengthen their drug loading and releasing capability. In general, MWCNTs are relatively superior to SWCNTs, as MWCNTs are cheaper to produce on a large scale, thus being more financially favorable for the drug delivery industry.

Since CNTs have a pre-formed supramolecular tube structure, the drugs can be loaded by CNTs through two approaches. The first approach is the capillarity-induced filling into the interior of CNTs. This mechanism is simple but the loadable amount of drugs is relatively low (below 5% (*w/w*)) [10]. The second approach is direct surface loading of drugs on CNTs via either possible covalent bonding or noncovalent interactions such as van der Waals contacts, hydrogen bonding, and π–π staking. However, covalent bonding is less favorable, since a minor alteration in a drug's molecular structure may change its effectiveness. For noncovalent adsorption, the aromatic ring on the surface of CNTs allows hydrophobic interactions to occur towards various drugs. This would especially favor most of the small-molecule drugs that contain flat benzene ring structures, and significantly improve their delivery efficiency. For instance, one investigation demonstrated that the loading capacity of aspirin could reach high as 48 wt % by suspending the functionalized MWCNTs in alcohol [11]. On the other hand, for certain drugs with a bulky structure, e.g., paclitaxel (PTX), their absorption is limited due to the lack of space on the surface of CNTs. Moreover, the resulting formulation is relatively unstable and the drugs adsorbed on the exterior of CNTs are more easily detached. One probable solution is to modify the surface of CNTs by incorporating hydrophilic or amphiphilic polymers to form micelles for encapsulating the drugs. These pre-functionalized polymer CNTs can provide expanded surface space, which would facilitate the molecular conjugation with bulk-volumed drugs and ligands. Shao et al. conjugated SWCNTs with a long-chain lipid docosanol molecule for the loading of PTX. A significant improvement of drug efficacy was achieved for both in vitro (78.5% for loaded drug vs. 31.6 for free drug, *p* < 0.01) and in vivo analysis using a human breast cancer xenograft mice model [12]. The formulated SWNTs-lipid-PTX was found to be non-toxic based on the test results of blood sampling and histological evaluation of major organs, though the raw CNTs have shown evident cytotoxicity. This would not only improve the biocompatibility of CNTs, but also favor the sustained release of drugs which are shielded by the three-dimensional polymeric space on the side walls of CNTs. Furthermore, after conjugating with the targeted drugs, the remaining surface sites of polymer CNTs are still free for linking to other functionalities, e.g., antibodies, fluorescence molecules, or even different type of drugs, for the purpose of multifunctional delivery [12]. A typical practice in this regard is to incorporate an additional anchoring moiety onto the drug-loaded nanoparticles, which can guide and tailor the drugs to a specific lesion [13].

Noticeably, many widely used antineoplastic chemotherapy drugs, such as DOX, cisplatin, and anthracyclines, require direct transportation into the nucleus. This is due to the nature of these drugs which are deoxyribonucleic acid (DNA) toxins and can generate free radicals to initiate apoptosis by inhibiting topoisomerase II. Therefore, a nuclear delivery

mode should be applied, and CNTs are particularly promising in this aspect because of their exceptional interaction mechanism with cellular membranes; they can penetrate the mammalian cell membrane via cytoplasmic translocation. This unique feature would potentially allow CNTs to directly deliver anticancer drugs into the targeted nucleus, instead of indirectly releasing the drugs in cytoplasm or lysosomes. However, the sidewall of CNTs is hydrophobic, which usually needs further chemical functionalization to improve the affinity with drugs and increase the in vivo aqueous solubility and biocompatibility of CNTs. Both covalent and non-covalent approaches have been conducted via the modification of different surface groups of CNTs, and their schemes are shown in Figure 2 [14].

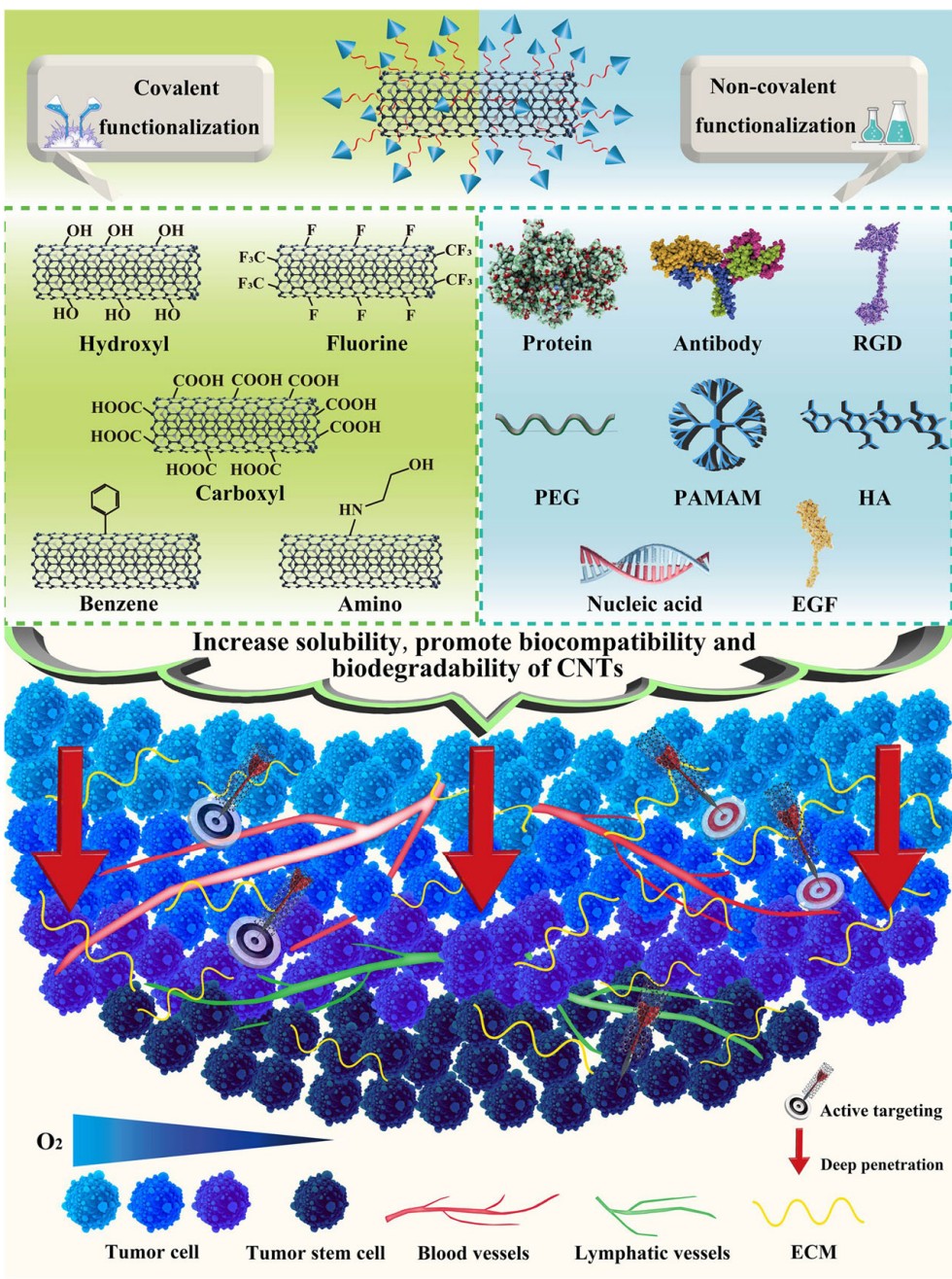

**Figure 2.** Schematic demonstration of various functionalization methods of CNTs via covalent and non-covalent approaches for active drug delivery and deep tumor penetration [14].

As a representative example, Liu et al. conjugated PEG-SWNTs with a cyclic arginine–glycine–aspartic acid (RGD) peptide. Efficient accumulation of the resulting SWNT composites at tumors was achieved in mice [15]. Pantarotto et al. studied the nuclear accumulation behavior of MWCNT nanocomposites, and revealed the ability of peptide-conjugated MWCNTs to penetrate the nuclear membrane [16]. This gives CNT nanocomposites potential to improve the cancer treatment modalities by minimizing the adverse effects of their delivered drugs. In fact, many trials have already been successfully conducted on the uptake of CNT nanocomposites as anticancer therapy. Tsai et al. synthesized a novel diblock polyglycolic acid (PGA)–co-heparin-conjugated MWCNT for improving the loading of DOX [17]. The hydrophilic heparin was selected not only due to its high nucleus sensitivity, but also because of its capacity to form an amphiphilic copolymer with PGA (PGA-Hep) to promote the dispersion of MWCNTs in fluid. This newly generated MWCNT nanocomposite is able to efficiently encapsulate the hydrophobic drugs via $\pi$–$\pi$ stacking interactions, followed by targeted drug release into the nucleus of HeLa cells. Heister et al. also designed a stable drug delivery system based on CNTs-PEG composites for effectively loading DOX and mitoxantrone, which featured sustained drug release and high selectivity against cancer cells [18]. After incubation for 72 h, 44% of DOX and 55% of mitoxantrone were released at pH 5.5, while at pH 7.4, only 7% of DOX and 8% of mitoxantrone were released.

## 3. Graphene/Graphite

Graphite is a naturally occurring allotrope of carbon. It can be formed via the reduction of sedimentary carbon during metamorphism. Graphene, on the other hand, is one single layer of graphite composed of $sp^2$-hybridized carbon atoms arranged in a honeycomb or hexagonal lattice. It has a two-dimensional structure with delocalized $\pi$ electrons on the aromatic rings, which can load drugs via $\pi$–$\pi$ stacking and electrostatic interactions [19]. Recently, both graphene and its derivatives have been explored as new and competitive nanocarriers for the delivery of therapeutic agents due to their intriguing physicochemical properties. Particularly, the excellent bio-functionalizability, selectivity, and solubility have largely expanded their applications in drug delivery, as well as other biomedical-related areas, e.g., cell culture and tissue bio-engineering.

As for drug loading and release specifically, the unique structural features, including surface area, layer number, lateral dimension, and surface chemistry, allow graphene and its derivatives to significantly improve the drug loading capacity with a tunable delivery profile. The specific surface area of graphene is around 2630 m$^2$/g, which is four magnitudes higher than other drug nanocarriers [20]. Because of its large surface area, graphene can provide multiple active sites with high drug loading capacity up to 200% (ratio of loaded drug weight to vehicle) [19]. The monolayer structure of graphene further improves the drug attachment since every atom could be exposed on surface for the interaction with drug molecules. As the number of layers of graphene sheets increases, their thickness and rigidity increase while lowering the surface area, thus negatively affecting the drug loading performance. The shape is also an important factor for graphene because of its unique two-dimensional structure with planar morphology, different from carbon nanotubes (tubular shape) and nanoparticles (spherical shape). In general, lateral dimensions have no substantial effect on drug loading, but could have size limitations relevant to biological degradation, cell uptake, renal clearance, and other biological activities dependent on particle dimensions [21]. Moreover, graphene can load drugs through both covalent bonds and physical adsorption, being more versatile than CNTs.

A wide range of drugs and bioactive compounds can be carried by graphene/graphite for creating sustained and targeted delivery systems. Table 1 summarizes a cluster of typical drug molecules which have been successfully loaded onto graphene. As can be seen, the majority of these drugs feature planar aromatic domains, which are able to form stable $\pi$–$\pi$ stacking, thus avoiding chemical conjugation.

**Table 1.** Drug molecules loaded onto graphene.

| Drug | Interaction | Functionalization | Tumor Type |
|---|---|---|---|
| Plasmid DNA | Electrostatic | PEI (electrostatic) | HeLa cervical carcinoma [22] |
| DOX | Hydrophobic | F127 (hydrophobic interactions) | MCF-7 breast cancer [23] |
| DOX and Camptothecin | π–π stacking/hydrophobic | FA (covalent) | MCF-7 breast cancer, A549 human lung carcinoma [24] |
| Photosensitizer (Chlorin e6) | π–π stacking | PEG (covalent) | KB nasopharyngeal carcinoma [25] |
| Ibuprofen and 5-fluorouracil | π–π stacking | Chitosan (covalent) | CEM human lymphoblastic leukemia [26] |
| siRNA | Electrostatic | PEG (covalent) | HeLa cervical carcinoma [27] |
| DOX | π–π stacking/hydrophobic interactions | PEG (covalent) | EMT6 murine tumor (in vivo) [28] |

Graphene oxide (GO) and reduced graphene oxide (rGO), as typical graphene derivatives, also offer a variety of functional groups, e.g., hydroxyl and carboxyl groups, for bioconjugation with drug molecules. The specific surface area of monolayer GO/rGO ranges from 2 to 1000 m$^2$/g, which is relatively smaller than pristine graphene but still has potential to enhance the drug loading [20]. Additionally, the photothermal properties of GO/rGO make it particularly applicable for targeted cancer therapy [29]. However, the hydrophobic nature and strong van der Waals forces between adjacent sheets make GO/rGO insoluble in water and difficult to disperse in most polar solvents, hindering its direct use as a drug adsorbent. In order to obtain a homogenous system, the negatively charged GO/rGO is usually conjugated with synthetic or naturally occurring polymers exhibiting positive charges. The generated dispersible complexes can form a stable framework with high porosity and enlarged specific surface area, which will then be able to adsorb various drugs with high efficiency. A typical configuration of a GO-based complex is shown in Figure 3, which was simulated by molecular dynamics (MD), representing the PEG-decorated GO (PEG-GO) loaded with DOX at human body temperature (310 K) and pH of 7.4 [30].

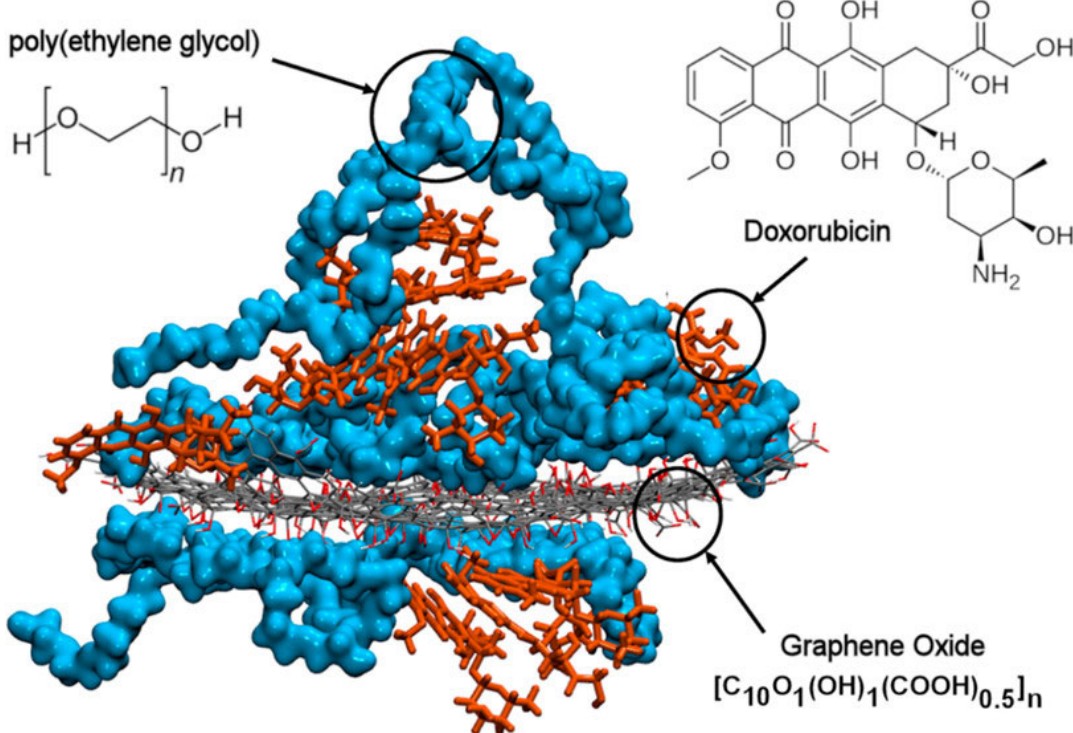

**Figure 3.** Molecular dynamics (MD) simulations of PEG-decorated GO (PEG-GO) nanocarriers loaded with DOX [30]. Reprinted (adapted) with permission from ACS Applied Bio Materials. Copyright {2020} American Chemical Society.

As for other trials, Pooresmaeil et al. grafted magnetic GO (MG) with β-cyclodextrin (β-CD) for the targeted delivery of DOX and methotrexate (MTX) as hydrophobic and hydrophilic anticancer drugs. High drug loading was achieved with approximate efficiency of 37.4% and 23.4% for DOX and MTX, respectively. The prepared β-CD-MG-drug nanocomposites exhibited better release behavior in cancer cells than in normal cells [31]. Tiwari et al. used similar functionalization strategies by conjugating polyvinylpyrrolidone (PVP) with GO (GO-PVP) and formulated a binary drug delivery system for the first time in cancer treatment. Two anticancer drugs, quercetin and gefitinib, were loaded onto GO-PVP with high efficiency of 20% for quercetin and 46% for gefitinib, which exhibited significant cytotoxicity against PA-1 ovarian cancer cells [32]. According to another study, the modified GO was even proposed as a carrier for quercetin via noncovalent interactions. The pristine GO was grafted with biocompatible hyperbranched polyglycerol (HPG) through the ring-opening polymerization of glycidol, which beneficially increased *d*-spacing between the basal planes. As a result, the dispersibility of GO was significantly enhanced. The prepared HPG-GO complex exhibited a high drug loading capacity of up to 185% and encapsulation efficiency of up to 93% while facilitating a sustained release of quercetin without initial burst in acidic solution. According to the in vitro release study, only 32.92%, 41.72%, and 49.22% of quercetin was released after 24 h, 48 h, and 72 h, respectively. Moreover, no evident cytotoxicity of the different concentrations of HPG-GO was observed on the MCF7 cell line during 72 h incubation [33].

Other graphene/graphite analogues, such as graphitic-phase carbon nitride (g-$C_3N_4$) and molybdenum disulfide ($MoS_2$), have been shown to be less toxic than GO and halogenated graphene, suggesting that they may be utilized for drug delivery [34,35]. While the as-prepared g-$C_3N_4$ or $MoS_2$ has superior features as a multifunctional nanomaterial, its low specific surface area largely suppress the drug loading capacity (e.g., the BET surface area of pristine bulk g-$C_3N_4$ was tested as only 56 $m^2$/g) [36]. As is known, the reaction between the drugs and g-$C_3N_4$ or $MoS_2$ initiates on the interface, so the efficiency largely depends on the specific surface area of g-$C_3N_4$ or $MoS_2$ particles. A higher surface area can provide more active sites to react with drug molecules, as well as improve the reaction kinetics. It is thus critical to increase the specific surface area of g-$C_3N_4$ or $MoS_2$, and a variety of strategies have been developed in this regard. Among such strategies, surface modification is considered as one of the most effective and facile approaches. Jiang et al. modified the g-$C_3N_4$ with a biomass-derived bio-oil which contained functional groups that are of electron-withdrawing character. The BET specific surface area of the modified g-$C_3N_4$ at reaction temperatures of 120 °C and 180 °C could reach as high as 172 $m^2$/g and 222 $m^2$/g, respectively. Correspondingly, their pore size (3.4 nm at 120 °C; 3.42 nm at 180 °C) and pore volume (0.3 $cm^3$/g at 120 °C; 0.34 $cm^3$/g at 180 °C) were also higher than the pristine g-$C_3N_4$ (3.05 nm; 0.11 $cm^3$/g). The drug loading amount can thus be increased, attributed to the synergistic effects of the enlarged specific surface area and enhanced porosity [36].

For $MoS_2$, Ferreira-Neto et al. prepared a new type of $MoS_2$ self-supported hybrid aerogel via assembling with bacterial nanocellulose (BC)-based organic macro/mesoporous scaffolds. The controlled and precise tuning of the synthetic parameters yielded a highly porous nanostructure with pore volume of 0.28–0.36 $cm^3$/g and interlayer distances of 0.62–1.05 nm. Together with its high surface area (97–137 $m^2$/g), these unique characteristics of the formulated BC/$MoS_2$ aerogel contributed to its exceptional adsorption capability [37]. In another study, a smart $MoS_2$-based nanoplatform was constructed for targeted drug delivery to human breast cancer cells. As shown in Figure 4, the poly(ethylene imine) and alpha-lipoic acid dispersed $MoS_2$ (PEI-LA-$MoS_2$) was functionalized with folic acid-grafted bovine serum albumin (FA-BSA), and then modified by PEG to enhance the dispersibility and colloidal stability. Its enriched loading of DOX could form a pH-sensitive system for shielding DOX in endosomes and releasing it to the cytoplasm in a controlled manner [38].

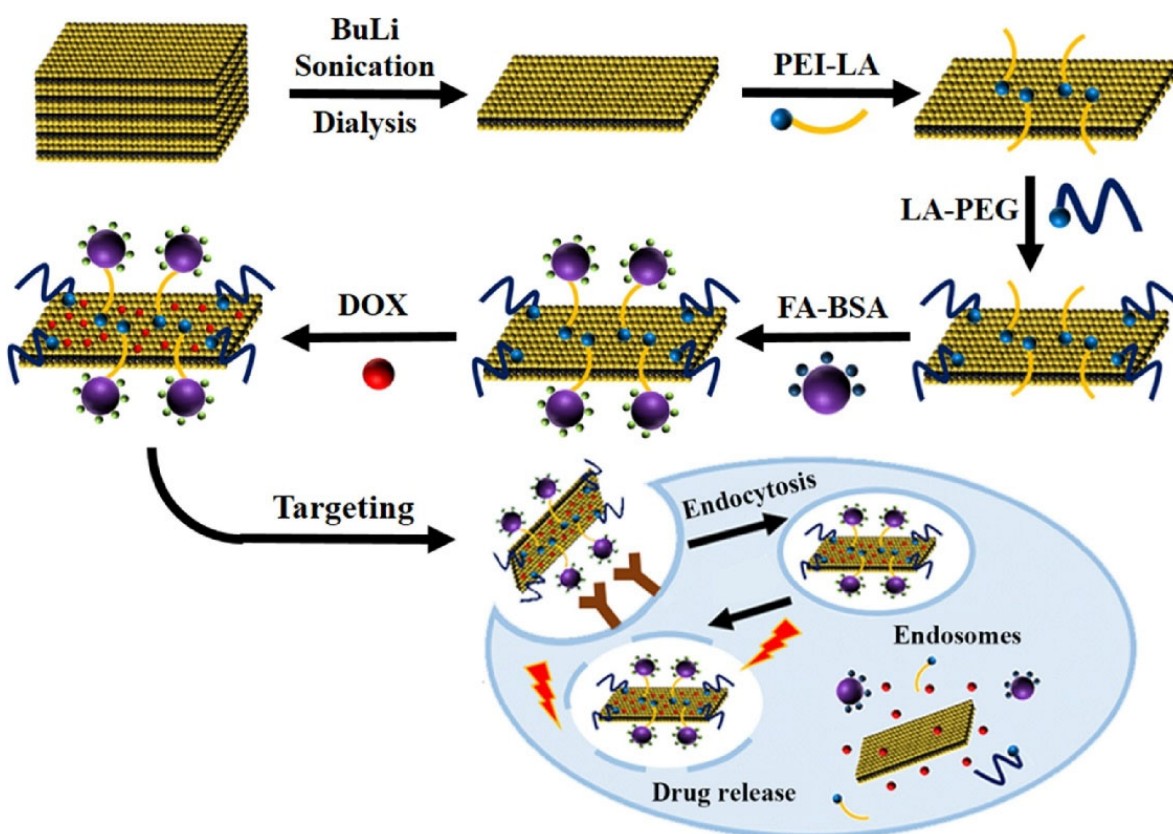

**Figure 4.** Schematic illustration depicting the construction of $MoS_2$-based nanoplatform for the enriched loading and targeted release of DOX [38].

In addition, the combination of $g-C_3N_4$ and $MoS_2$ in their improvement of drug adsorption has also been investigated. Nouri et al. loaded FA and DOX into the $g-C_3N_4/MoS_2$ incorporated-chitosan/ethyl cellulose nanofibers for the targeted drug delivery against HeLa and MCF-7 cancer cell lines. The pharmacokinetic studies of this novel $g-C_3N_4/MoS_2$-based carrier revealed its linear and sustained release of FA and DOX with non-Fickian diffusion. High anticancer activity against HeLa and MCF-7 was achieved with in vitro cell death ratios of 85% and 89%, respectively, after 7 days of the treatment period [39].

## 4. Fullerene

Fullerene has been shown to be potentially useful as a drug absorbent due to its intrinsic apolar character. It is a unique class of carbon allotropes with a representative molecular structure of $C_{60}$, and individual carbon atoms are connected with one another by two types of C-C bonds: one shared by two neighboring hexagons ($R_{66}$), and the other by a hexagon and pentagon ($R_{65}$) [40]. The optimized $R_{66}$ and $R_{65}$ bond distances in the fullerene structure are 1.40 Å and 1.45 Å, respectively, which differ from the C-C bond distance in pristine graphene of about 1.42 Å [41]. This unique chemical configuration structured a fullerene-based composite with a highly porous architecture, which is able to remarkably enhance its drug delivery profile.

Since its discovery, fullerene has been widely explored in the pharmaceutical field due to its intrinsically appealing electrochemical and physical properties. However, similar to graphene, the direct use of virgin fullerene in drug delivery is limited because of its poor solubility. In general, this issue can be addressed by a suitable chemical modification, but with the risk of damaging the intrinsic physiochemical properties of fullerene. Alternatively, the complexing of fullerene with polymers was found to be more practical, and is thus being widely explored. The resulting fullerene nanocomposites can function as efficient vehicles

for drug delivery with high loading capacity. Shi et al. encapsulated polyethylenimine-derivatized fullerene (PEI-fullerene) with FA for the conjugation of docetaxel (DTX). The resulting PEI-fullerene-FA/DTX showed an increased in vitro antitumor efficacy with a 7.5-fold higher DTX uptake in cultured PC3 cells compared with free DTX [42]. In addition, considering that PEI is cytotoxic at low to medium molecular weight owing to its cationic nature and non-cleavable molecular structure, the combination of PEI's amino groups with polymeric moieties, such as fullerene-FA, could also alleviate its toxicity to normal organs. This is in agreement with the findings reported by Uritu et al., which showed non-cytotoxicity of the linear or branched PEI-fullerene-polyethylene glycol dendrimeric structures [43].

Recently, the biopolymer-based fullerene nanocomposites are attracting intensive attention for delivering drugs owing to their unique eco-friendliness and biocompatibility [44]. These nanocomposites can combine the advantages of both components (fullerene and biopolymer) and show some new and synergistic capability to be used as nanocarriers. Tan et al. prepared the amphiphilic block copolymer/fullerene ($C_{60}$) micelles for the delivery of DOX, by incorporating fullerene into the hydrophobic core of methoxy polyethylene glycol-poly(d,l-lactic acid) (MPEG-PDLLA). It was found that the incorporation of the intact spherical fullerene with MPEG-PDLLA favored the dispersion of fullerene in physiological media and increased the molecular chain space of PDLLA segments in the vicinity of fullerene. This could provide a larger cargo space for enhancing the drug entrapment [45].

Another concern regarding the application of fullerene is that though its electronic charges are uniformly distributed within the entire symmetry structure, the density functional theory (DFT) calculations revealed that the above of the center of pentagon and hexagon rings are the most active adsorbing sites [46]. Various strategies have been explored in modifying its adsorption homogeneity, and manipulating fullerene with dopants is one of the most effective approaches [47]. The doping of additives/impurities can offer additional sites for fullerene to interact with drugs via noncovalent binding. For example, a recent quantum mechanics study revealed that doping fullerene with a B atom can transform this inert material into an active nanocarrier for the delivery of aspirin [48]. Srinivasu et al. prepared a porous fullerene, $C_{24}N_{24}$, by truncated doping of 24 carbon atoms with 24 nitrogen atoms. The six generated $N_4$ cavities showed high binding energies with transition metal atoms, including Sc, Ti, and V (nearly double that of the corresponding metal cohesive energies). A high adsorption capacity of molecular hydrogen was achieved with the calculated adsorption energies of −9.0 to −3.0 kcal/mol [49].

The direct doping of transition metals has also been investigated to trigger and regulate the adsorption behavior of fullerene. Alipour et al. studied the interaction between porphyrin-like metal-doped fullerenes with a non-steroidal anti-inflammatory drug, ibuprofen (Ibp). As shown in Figure 5, the doped fullerene with transition metals ($TMN_4C_{55}$, TM = Fe, Co, and Ni) possessed higher adsorption energies and shorter interaction distances with Ibp compared with pristine fullerene ($C_{60}$) ($E_{ads}$ = −13.14 kcal/mol, d1 = 3.24 Å, d2 = 2.67 Å), which indicated the enhanced chemisorption between Ibp and $TMN_4C_{55}$. Among all the dopants, Ni exhibited the highest adsorption energy due to its shortest interaction distance with the O atom from Ibp ($E_{ads}$ = −23.11 kcal/mol, d1 = 1.96 Å), and thus was considered as the most favorable loading site for Ibp [40]. More interestingly, instead of using transition metals, Esrafili et al. decorated fullerene with alkali metals (AM = Li, Na, and K) for the first time in the delivery of 5-fluorouracil (5FU). The water-soluble 5FU is an antimetabolite chemotherapeutic drug which has been widely used in cancer treatment, but with the drawback of poor bioavailability. By adsorbing onto the fullerene/AM system, its therapeutic efficacy was significantly improved through boosting the cellular uptake, with minimized adverse effects. The adsorption energies of 5FU with fullerene/AM were −19.33, −16.58, and −14.07 kcal/mol for Li, Na, and K, respectively, which means up to twelve Li atoms, six Na atoms, or one K atom could be anchored onto fullerene's surface. Simultaneously, each of these AM atoms could carry one 5FU molecule. In addition to the multiplied loading capacity, the fullerene/AM system could easily release 5FU molecules

when it reaches the targeted cancerous tissues, due to the moderate adsorption energies and charge transfer values between 5FU and fullerene/AM [46].

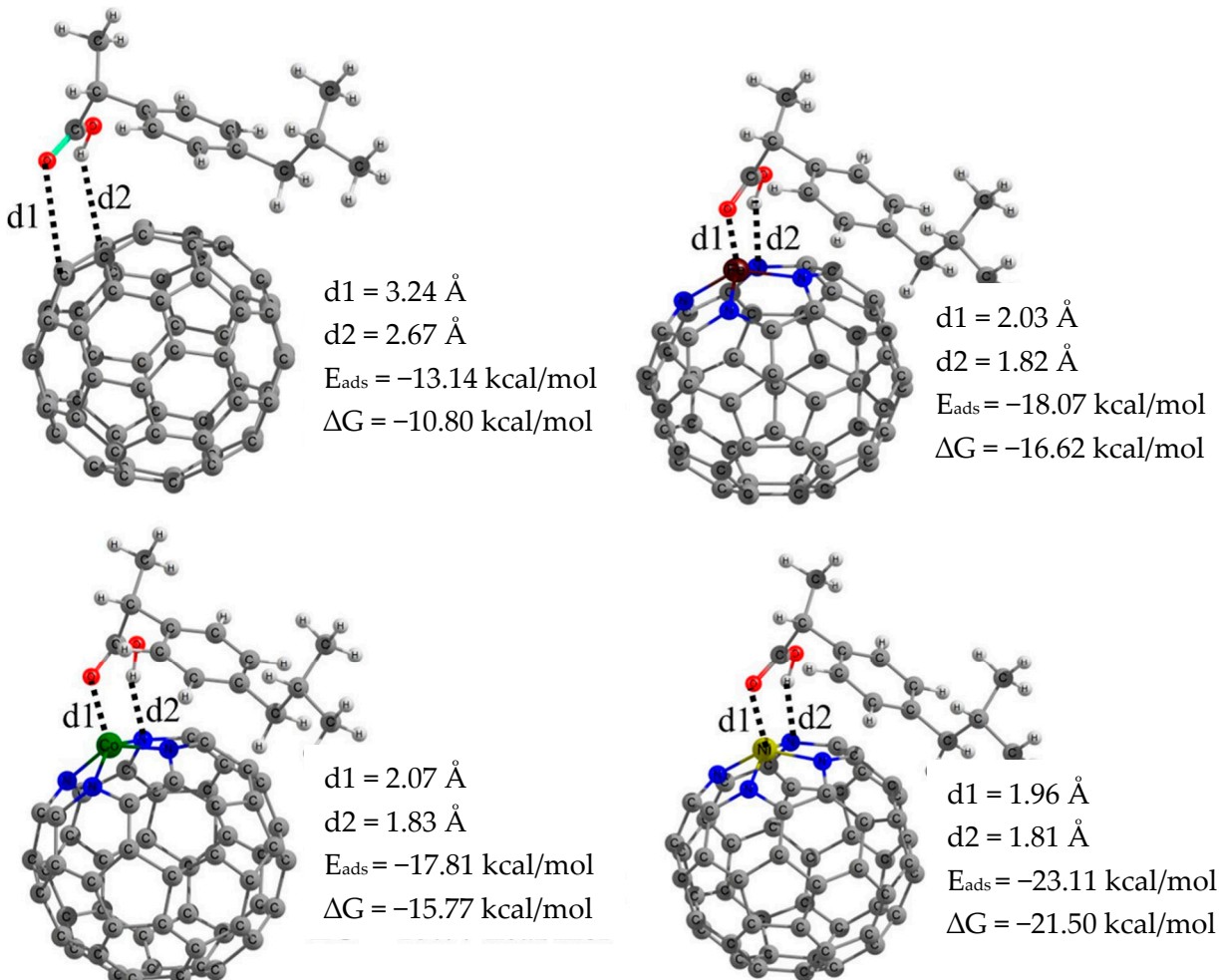

**Figure 5.** Molecular configurations of the adsorbed Ibp on the surface of doped fullerenes [40].

For cancer treatment, fullerene is particularly promising due to its unique spherical structure, a truncated icosahedron, along with non-dipole character. A wide range of anticancer therapeutics from antibodies, proteins, and genes to small drug molecules can be carried by fullerene for creating sustainable and targeted delivery systems. Since most of anticancer drugs have low solubility in an aqueous environment, the functionalized fullerene has excellent water dispersibility to stabilize these hydrophobic drugs, without requiring surfactants or additional oxidation, as seen for other graphene-based nanomaterials. Maleki et al. investigated the interactions between functionalized fullerene and DOX/Paclitaxed (PAX) by molecular dynamics (MD) simulations. The results showed that functionalizing fullerene with carboxyl groups could enhance its loading and releasing capacity for both DOX and PAX. However, DOX exhibited a better interaction than PAX, based on the results of electrostatic and Van der Waals interactions analysis. The latter could be improved by grafting the trimethyl chitosan (TMC) polymer with fullerene, attributed to the significantly increased number of hydrogen bonds between PAX and fullerene at neutral pH [50].

## 5. Nanodiamonds/Diamond-like Carbon (NDs/DLC)

Over the past decade, nanodiamonds (NDs) and diamond-like carbon (DLC) have attracted tremendous interest in the biomedical field for high-loading drug delivery. NDs

refer to the type of diamond particles with sizes from a few to nearly 100 nanometers, and shown in Figure 6 is a representative ultrasmall 5 nm detonation ND and [1(2,3)4]pentamantane, $C_{26}H_{32}$, in comparison with one of the largest synthesized molecules, 7.7 nm diameter tetracosakis([2-hydroxy-5-(octyloxy)-1,3-phenylene]dimethylidene)dodecakis(5,10,15,20-tetrakis(4-aminophenyl)porphyrin), $C_{912}H_{841}N_{96}O_{48}$, in an atomistic scale [51]. NDs are known as ultradisperse crystals with exclusive properties of chemical inert core (sp$^3$ carbon atoms) and an amorphous shell with hanging bonds ended of functional groups for the interaction with therapeutic drugs [52]. These groups can also be easily derivatized with secondary functionalities, and the resulting ND derivatives provide a versatile platform for conjugation with an even broader range of drugs [53].

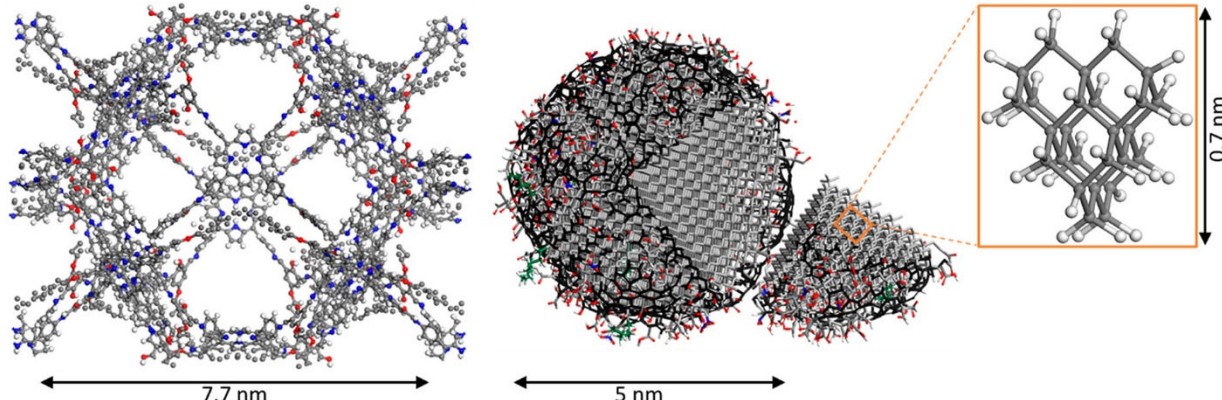

**Figure 6.** Optimized atomistic models: (**left**) one of the largest synthesized molecules, 7.7 nm tetracosakis([2-hydroxy-5-(octyloxy)-1,3-phenylene]dimethylidene)dodecakis(5,10,15,20-tetrakis(4-aminophenyl)porphyrin), $C_{912}H_{841}N_{96}O_{48}$; (**middle**) 5 nm DND with diamond core structure and different types of functional groups and sp$^2$ C on the surface; (**right**) [1(2,3)4]pentamantane, $C_{26}H_{32}$, a representative diamondoid [51].

To take further advantage of NDs' potential in drug delivery, recent focus has also been laid on their purity and other factors which may directly or indirectly affect the drug adsorption and release in the biological environment. In general, the purified NDs would have an almost perfect crystalline structure centered with naturally occurring nitrogen vacancy (N-V) or nitrogen impurities, which can form complexes such as peptides or amines [53,54]. This would allow them to act as a flexible template around the curved surface where electrons are unstable, and the formulated truncated octahedral architecture is able to further enable the potent drug loading [55].

Diamond-like carbon (DLC) is a type of amorphous carbonaceous material with a hybrid formation of graphite (sp$^2$-bonded carbon) and diamond (sp$^3$-bonded carbon) in their overall structure. Its physiochemical properties are mainly determined by the ratio of sp$^2$/sp$^3$, along with the surface functionalities. This distinctive nature has branded DLC as an efficient nanocarrier for the conjugation of active drug molecules. In particular, its large surface area and unique diamond-like structure can provide more binding sites, thus fundamentally improving drug loading. The formulated DLC–drug complex can be presented in many forms, e.g., a thin DLC film which interacts with a drug in two dimensions, or a spontaneous DLC–drug complex hydrogel with low free energy [56]. Apart from loading hydrophilic drugs, DLC has also emerged as an efficient dispersibility-enhancing agent for poorly water-soluble drugs [57]. This characteristic is attributed to DLC's capability of adsorbing drugs on surfaces while maintaining their therapeutic effectiveness. Its abundant surface groups, e.g., phenols, pyrones, and sulfonic acid groups, are able to stabilize DLC in its suspension form, which allows bulky attaching of water-insoluble drugs such as the anti-inflammatory drug dexamethasone, or anticancer drug 4-hydroxytamoxifen (for breast cancer) [54]. Moreover, for the majority of protein drugs,

DLC is able to immobilize them with less distortion of their conformation, and allows better binding of antibodies to create a stronger immune response [58].

NDs/DLC can be easily complexed with drug compounds, and their properties of high colloidal stability in aqueous and non-aqueous solution allow NDs/DLC to be efficiently used in drug loading [59]. Various studies have been carried out in this regard to address the delivery of drugs to biological targets both in vitro and in vivo. Enomoto et al. reported a drug eluting stent (DES) system by coating the biocompatible polymers with micro-patterned lattice-like DLC. This newly designed DES system possessed sufficient antithrombogenicity and could inhibit the initial drug burst release, thus being considered as a competent candidate for the treatment of patients with obstructive coronary artery disease. By further manipulating the coating area of DLS on the polymer surface, the drug eluting profiles could be effectively controlled, creating potential for the new generation of DES with a possible complete prevention of late thrombosis [60]. In the field of ocular drug delivery, NDs have been successfully used for the treatment of glaucoma. The diamond particles were first coated by polyethyleneimine (PEI), and then cross-linked with chitosan to form ND nanogels for the loading of timolol maleate (TM). These ND nanogels were cast into contact lenses after their embedding within the poly-2-hydroxyethyl methacrylate (poly-HEMA) matrix. Effective complexation and release of the drug was achieved [61].

Moreover, NDs/DLC have the potential to deliver drugs into a specific single cell, and thus are suitable for cancer treatment, which usually requires targeted drug delivery to cancer cells [62]. In fact, an ND/DLC-based delivery system against cancers has been developed as one of its most promising biomedical applications. Plus, certain distinctive types of diamond nanoparticles can be uptaken by living cells via the mechanism of clathrin-dependent endocytosis; interestingly, the amount of nanodiamonds uptaken by cancer cells quantitatively exceeds that uptaken by non-cancer cells, which can further promote the use of nanodiamonds in anticancer therapy. Thus far, a series of studies have been documented regarding the use of drug-loaded NDs/DLC in cancer treatment. Toh et al. synthesized an ND–mitoxantrone (MTX) complex, which exhibited a marked improvement in its therapeutic efficacy against cancer cells [63]. The hitherto findings indicate that ND/DLC-mediated drug delivery is able to serve as a powerful system to overcome the chemoresistance in cancer stem cells and significantly increase the treatment efficiency.

Another investigated approach for improving the drug loading efficacy was to reversibly bind NDs/DLC to therapeutic drugs via ionic interactions. This type of pseudo-coating mechanism was able to prevent the leaching of toxic ions from certain drugs into the body, thus minimizing the drug toxicity. Huang et al. formulated an ND–doxorubicin hydrochloride (DOX·HCl) cluster through the interaction between hydroxylic/carboxylic groups of DLC and amine groups of DOX·HCl. The resulting NDs-DOX·HCl cluster comprised a loose structure, which could provide additional spaces for DOX·HCl to be adsorbed both on the ND surface and in the fissures of the cluster. This unique hybrid formulation exhibited lower cytotoxicity than free DOX·HCl, because the clusters of NDs surrounding the drugs made them unapproachable by healthy cells, and they remained intact until they reached the targets. Therefore, it enabled most of the DOX·HCl drugs to release at the designated single cell or lesion sites. The lowered cytotoxicity was confirmed by the results of cytotoxic studies on mouse macrophages and human colorectal cancer cells [64]. Upon completion of treatment, the residual diamond nanoparticles can be easily excreted from the body by the kidneys without blocking blood vessels, which is beneficial in further diminishing the adverse effects of the NDs/DLC–drug cluster [65]. However, there is a common concern regarding the inconstant stability of the aforementioned clusters/complexes, since it is dependent on the bonding of NDs/DLC with respective drugs. In general, a cluster/complex generated by covalent bonds is favorable, since it is more stable than the one formulated by non-specific absorption [66].

## 6. Challenges towards Carbonaceous Nanocomposites: Toxicity

Despite their superb performance in enhancing drug delivery, carbonaceous nanocomposites also have certain limitations in their biomedical applications, among which toxicity remains the major one. (1) During the manufacturing process, the low-density carbonaceous powder may spread into the air and cause pulmonary toxicity through respiration [54]. (2) Due to the ultrasmall size of carbonaceous carriers, an expensive radionuclide tracer technique is usually applied to detect these nanoparticles [67]. This type of technique is able to accurately evaluate the distribution of nanoparticles, however, it involves the use of radioactive substances, which can lead to toxicity. (3) Cytotoxicity due to the aggregation or flocculation of carbonaceous carriers on cell membranes in the body is another limitation of carbonaceous nanocomposites.

In general, the interactions between carbon particles with cells/macrophages depend on several factors, including the lateral dimensions, shape, and surface chemistry, which could have a significant impact on toxicity. Rigidity is another important parameter in maintaining the structural integrity of carbonaceous nanocarriers, but could also cause cell damage if their structure is too rigid. It thus requires an optimized level of rigidity of carbonaceous nanocomposites, which creates another obstacle for their drug delivery application [21].

One intensively reported method of alleviating the toxicity of carbonaceous nanocomposites was to modify their surface chemistry, with the aim to improve their biocompatibility with cells. However, there are some concerns regarding the involved modification process, such as the complicated multi-step procedure, high temperature above 100 °C, and time consumption. Moreover, the synthesis of carbonaceous nanocarriers with active moieties by covalent bonding usually uses a corrosive solution, which may lead to additional toxicity [19]. Therefore, this issue remains relevant to the development of carbonaceous drug nanocarriers moving forward.

## 7. Carbon-Based Nanocellulose

In order to address the aforementioned technical challenges, pioneering research has been focused on biomass-based carbonaceous nanocomposites. As is known, nature-resourced materials often possess a hierarchically organized structure with a certain periodic pattern, exhibiting potential to be converted into advanced carbon derivatives with desired properties for engineering composites. Nanocellulose, as one typical carbon-based material, can be considered as an ideal carrier for drug delivery, with the aim of minimizing the side effects due to its known non-immunogenicity [68,69]. Moreover, the high surface-to-volume ratio, superior thermal stability, and facile functionalization give nanocellulose the potential to enhance its drug delivery capacity [70–72].

The basic structure of nanocellulose consists of repeating anhydro-D-glucopyranose units (AGU) bonded by β(1→4) glycosidic linkages. Each AGU contains three hydroxyl groups (Figure 7a) [73], which possess superior affinity towards various drugs. Other basic functionalities such as carboxyl groups can provide additional control of drug release in the intestinal environment. Meanwhile, the negatively charged sulfate groups allow an isotropic and stable distribution of drug-loaded nanocellulose in the solvent medium. Regarding the factor of morphology, rod-like nanocellulose favors adsorption of drugs by intermolecular interaction with the drug surface. Shown in Figure 7b is the TEM image of nanocellulose particles with width of 3–10 nm and length of 50–165 nm [74]. The small size of these nanoparticles would allow them to move more easily in the body than larger materials, and thus may avoid the burst local release of drugs.

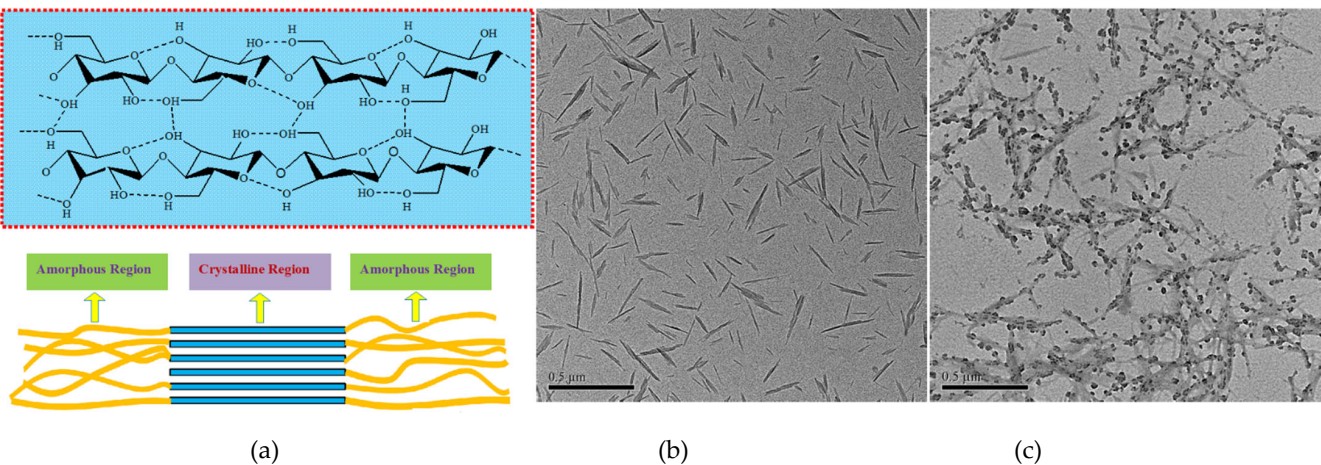

**Figure 7.** Molecular structure of nanocellulose macromolecule (**a**); TEM image of nanocellulose (**b**); nanocellulose–SO compound (**c**) [75,76].

Nanocellulose has the capacity to attach different molecules through electrostatic interactions. Shown in Figure 7c are the nanoparticles of Spirooxazine (SO)-loaded nanocellulose, which exert higher photochromic efficiency as well as improved color stability and fatigue resistance [75,76]. As for therapeutic drug delivery, various water-soluble drugs can bind and release from nanocellulose owing to its abundant surface functionalities. Their capacity for further modification can largely broaden the spectrum of drugs, e.g., hydrophobic drugs, capable of binding to nanocellulose. The final product can be fabricated in the form of capsules, films, microparticles, and gels for topical, oral, and transdermal administration [68]. For instance, Thomas et al. successfully designed an oral formulation of alginate–nanocellulose hybrid (ALG-CNC) for the controlled delivery of rifampicin (RIF). A high drug entrapment efficiency of 69.73% was achieved, attributed to the small average size of synthesized ALG-CNC (70 nm). The addition of nanocellulose can overcome the limitations of ALG, e.g., low mechanical strength and poor porosity, without disturbing its inherent network structure. The ALG-CNC hybrid showed improved stability with low swelling at pH 1.2, which could prevent the burst drug release in harsh gastric conditions [77]. As for topical use, one of the representative examples was reported by Sun et al., who showcased an innovative hand sanitizer formulated with antimicrobial intensively loaded nanocellulose (Figure 8) [78–80]. The well-distributed drug particles (modified triclosan) on the surface of nanocellulose exhibited enhanced efficiency of germ killing, which has been playing an important role in tackling COVID-19. This newly developed alcohol-free disinfectant product can provide persistent protection without causing skin dehydration. Moreover, nanocellulose exhibits no photo-catalytic activity, and thus can prevent the generation of reactive oxygen species (ROS) in causing sun damage of skin. Plus, the nanocellulose in its film form can scatter UV light, which can shield the skin from UV-B radiation. Uniquely, as one typical example of exploring the synergistic effect between different carbonaceous nanocarriers, nanocellulose was also applied to attach CNTs for facilitating their even distribution [81,82]. The electron microscopy analysis revealed that nanocellulose particles could be well anchored onto the tips of CNTs, thus orienting the nanotubes in the same direction. The improved distribution can potentially increase the drug loading efficiency via the superimposed functioning of nanocellulose and CNTs.

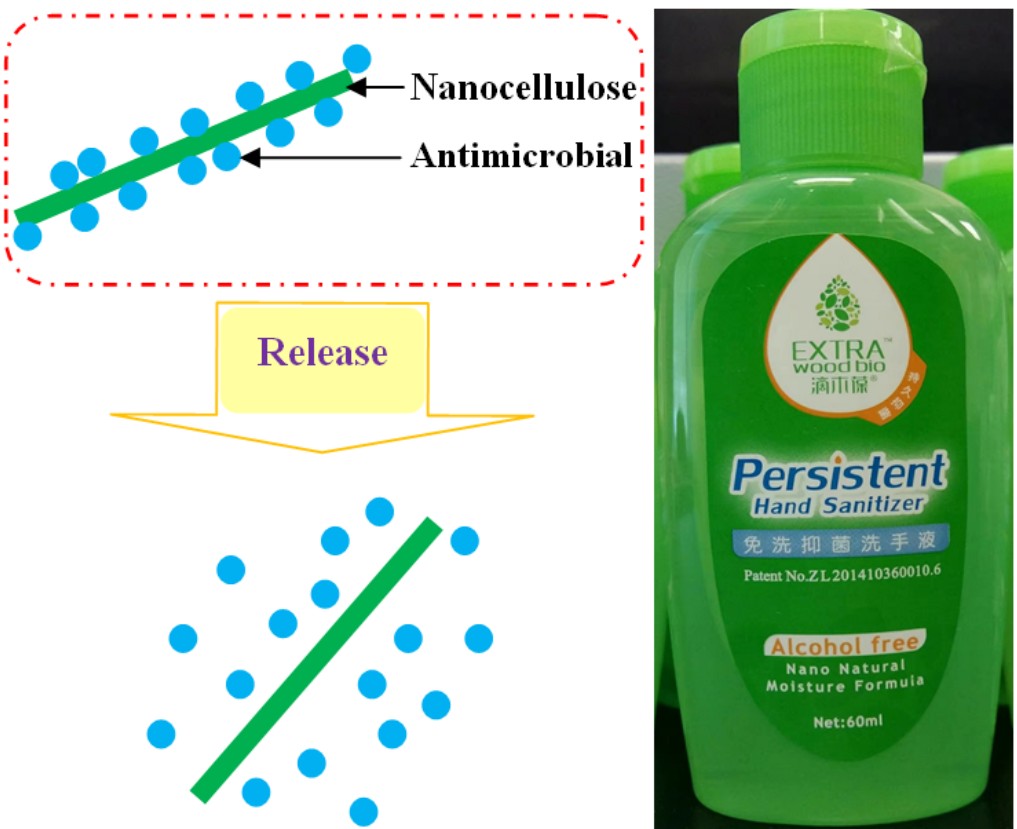

**Figure 8.** Loading and release mechanism of the nanocellulose-formulated hand sanitizer.

Other types of nanocellulose, e.g., cellulose nanofibers (CNFs) or bacterial cellulose (BC), have also shown potential as carriers for drug delivery [83]. In 2019, Plappert et al. developed a transdermal drug delivery patch by assembling CNFs into anisotropic-layer membrane for the delivery of piroxicam. The high surface area and tunable carboxylate content of the CNF membrane significantly increased its adsorptive affinity with piroxicam and enhanced the drug solubility. A prolonged in vitro release of piroxicam was realized under the simulated human skin conditions [84]. Moreover, electrospun nanofibers, with tailorable properties by easily controlling the process parameters, have recently been applied for improving drug loading. The treatment of disease and infections with electrospun nanofiber–drug complex has the benefit of increasing the drug concentrations at local sites, thus minimizing the drug's adverse effects. In previous studies, DOX-loaded electrospun nanofibers were used for drug delivery, which exhibited high efficiency against cancer cells. Similar results were reported with other drugs, such as paclitaxel, cisplatin, and dichloroacetate, in improving their anticancer capacity [85].

As for bacterial cellulose (BC), it can be mixed with other pharmaceutical excipients, and the addition of BC offers several benefits, such as increased dissolution rate, the potential of gastroretentive delivery due to the induction of positive buoyancy, and the sustained drug release in fasted state-simulated stomach. Uniquely, the similarity in the nanostructure and collagen morphology makes BC especially suitable as a drug carrier for cell immobilization and cancer therapy [86]. One previous trial using BC in localized cancer treatment for controlled release of DOX was reported by Cacicedo et al. The purpose of their study was to maximize the drug accumulation at tumor sites and to eradicate the side effects of administered DOX. Two sets of formulations, nanostructured lipid carriers (NLCs) containing cationic DOX (NLCs-H) and neutral DOX (NLCs-N), were encapsulated into a BC matrix (BC-NLCs-NH). A higher encapsulation efficiency (97%) and sustained drug release were reported with NLCs-H, and the synthesized BC-NLC-NH composites could significantly decrease the tumor-to-control (T/C) ratio of ex vivo tumor volume

(53%) and tumor weight (62%). No evident side effects, such as edema, inflammation, or necrosis, were observed, which demonstrated the suitable biocompatibility of the BC delivery system [87].

To date, the development of high-performance drug nanocarriers derived from nanocellulose composites has been considered as a focal point in the modern biomedical field. However, there are also several concerns regarding the adverse effects from the possible over-accumulated nanocellulose particles for in vivo applications [86,88]. Therefore, an in-depth understanding of how nanocellulose's nature may affect the living cells is necessary.

## 8. Conclusions

Carbonaceous nanocomposite-mediated drug delivery has drawn significant attention for the reliable transportation of active pharmaceutical ingredients in biomedical applications. Their tunable structural properties and facile chemical modification can significantly boost drug loading and enable sustained release in living systems. The accompanied advantages upon utilizing carbonaceous nanocomposites include increasing the drug stability, controlling the drug dissolution rate, extending the cycle duration, and minimizing the adverse effects, all of which can improve the drug administration safety and efficacy. This has led to a rapid development of chemotherapeutics therapies in promoting the use of carbonaceous nanocomposites. Still, many challenges such as the undesired toxicity remain issues of high concern. As a result, advanced knowledge regarding the structure and surface chemistry of carbonaceous nanocomposites has been continuously explored to better control their properties to meet the demands from consumers for green and healthy living. Particularly, the carbon-based nanocellulose possesses unusual and desirable properties, such as higher biocompatibility, uniform nanorod shape, and unique liquid crystalline character, in comparison to other carbon bulk materials, which have promoted its increased use in newly developed drug nanocarrier systems. However, there is a possibility that nanocellulose might over-accumulate in the body. Therefore, nanocellulose designated for drug delivery must be thoroughly assessed for potential health risks before further recommendations can be made on its large-scale application, especially related to transdermal and oral administration.

**Author Contributions:** Conceptualization, B.S. and M.S.; methodology, B.S.; software, B.S.; validation, W.W.; formal analysis, M.S.; investigation, M.S.; resources, W.W.; data curation, B.S.; writing—original draft preparation, B.S.; writing—review and editing, W.W. and M.S.; visualization, W.W.; supervision, M.S.; project administration, M.S.; funding acquisition, B.S. and M.S. All authors have read and agreed to the published version of the manuscript.

**Funding:** This research was funded by the National Natural Science Foundation of China, grant number 31501440; the Hebei Provincial Scientific and Technological Cooperation & Development Foundation between Province and University of 2018; the Tianjin Science and Technology Commissioner Program, grant number 16JCTPJC45300; the Tianjin International Training Program for Excellent Postdoctoral Fellows of 2015; and the China Postdoctoral Science Foundation, grant number 2015M571268. The authors also acknowledge NSERC of Canada, Ontario Research Fund: Research Excellence, Ford Canada, and Total NA for continuing support of this work.

**Conflicts of Interest:** The authors declare no conflict of interest.

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
