# Peer review of "Carbonaceous Nanocomposites for Biomedical Applications as High-Drug Loading Nanocarriers for Sustained Delivery: A Review"

_jcs, doi:10.3390/jcs6120379_

Round 1

Reviewer 1 Report

Summary this is a review of different technologies and techniques such as carbonaceous nanocomposites used to improve drug loading capacity being greater than 10% from polymeric nanoparticles.  

Overall the paper is well written and I enjoyed reading it. I have only a few comments for the authors. 

Page 3: 

Lines 112 – 118: Do the large molecules being added to the CNTs block other therapeautic drugs for multifunctional delivery or are the CNTs long enough to prevent steric hindrance of drugs keeping other drugs from binding to the CNTs? 

Page 5:  

Line 51 - 53: What is considered a sustained drug release for CNTs? For example, most nanoparticles release drug within a few days and in some instances months, and topical drugs on skin release within hours.  

Page 7: 

LInes 23 – 27: These drugs are being loaded to higher efficiencies with HPG-GO but what is their drug release profile? 

Is there any insight into why GO is highly biocompatible such as the oxide film and polyglycerol make the composite more hydrophilic thus making the polymer less cytotoxic? 

Lines 49 – 60: Does the porosity increase of BC lead to a increased burst release of DOX? Are there any graphs of this drug release to add to paper? 

Page 12: 

LIne 27 – 29: How does a controlled release of a drug lead to prevention of late stent thrombosis? Late stent thrombosis is usually a result of the drug no longer being released leading to inadequate tissue coverage of the stent.  

Page 13:  

Lines 68 – 70: Do carbonaceous nanocomposites degrade and how are they eliminated from the body if implanted?

Reviewer 2 Report

This review summarizes reports on the use of nanocarbons and nanocelluloses as carriers for drug delivery. The manuscript is well-written. I recommend its publication with the following corrections:

1.     What is the uniqueness of this review compared with previously reported reviews? There are a number of reviews on nanocarbons and nanocelluloses for biomedical applications, including drug delivery.

2.     Line 78: “soluble” should not be used for colloidal particles including CNTs. Instead, “dispersible” is appropriate.

3.     Lines 78–80: Why MWCNTs are partially water-dispersible even though they are hydrophobic as with SWCNTs?

4.     Lines 279–280: What is the meaning of “This unique chemical configuration structured fullerene with highly porous architecture” ?; fullerene is spherical and is not porous.

5.     Line 537: What is the antimicrobial drug shown in Figure 8?

6.     Lines 544–550: It appears that the paper on the use of nanocellulose for distributing CNTs is not cited.

7.     Lines 568–597: Why this paragraph is in the section “7. Carbon-based Nanocellulose” even though it does not include nanocellulose? This paragraph should be removed from this section.

8.     Line 609: What is “NLCs” ?

Round 2

Reviewer 1 Report

The paper reads well.

Reviewer 2 Report

The authors have addressed reviewer's comments. I recommend publication of the current manuscript.